# Oxygen-Sensitive Photo- and Radioluminescent Polyurethane Nanoparticles Modified with Octahedral Iodide Tungsten Clusters

**DOI:** 10.3390/nano12203580

**Published:** 2022-10-12

**Authors:** Vyacheslav A. Bardin, Yuri A. Vorotnikov, Dmitri V. Stass, Natalya A. Vorotnikova, Michael A. Shestopalov

**Affiliations:** 1Nikolaev Institute of Inorganic Chemistry SB RAS, 3 Acad. Lavrentiev Ave., 630090 Novosibirsk, Russia; 2Voevodsky Institute of Chemical Kinetics and Combustion SB RAS, 3 Institutskaya St., 630090 Novosibirsk, Russia; 3Department of Physics, Novosibirsk State University, 2 Pirogova St., 630090 Novosibirsk, Russia

**Keywords:** octahedral halide tungsten cluster, polyurethane nanoparticles, photoluminescence, radioluminescence, singlet oxygen generation

## Abstract

The development of cancer treatment techniques able to cure tumors located deep in the body is an urgent task for scientists and physicians. One of the most promising methods is X-ray-induced photodynamic therapy (X-PDT), since X-rays have unlimited penetration through tissues. In this work, octahedral iodide tungsten clusters, combining the properties of a scintillator and photosensitizer, are considered as a key component of nanosized polyurethane (pU) particles in the production of materials promising for X-PDT. Cluster-containing pU nanoparticles obtained here demonstrate bright photo- and X-ray-induced emission in both solid and water dispersion, great efficiency in the generation of singlet oxygen, and high sensitivity regarding photoluminescence intensity in relation to oxygen concentration. Additionally, incorporation of the cluster complex into the pU matrix greatly increases its stability against hydrolysis in water and under X-rays.

## 1. Introduction

Photodynamic therapy (PDT) is a very promising but still emerging method of non-invasive cancer treatment [1,2]. In general, the photodynamic effect is a generation of reactive oxygen species (ROS) in the presence of photosensitizers (PSs). There are two types of PSs (I and II), each with different ways of generating ROS [3]. Excited type I PSs react with substrate, which results in the formation of different radicals and radical ions, while excited type II PSs transfer energy to triplet oxygen forming singlet oxygen (^1^O_2_). In conventional PDT, PSs are mostly excited with visible or near infra-red (NIR) light; however, NIR light has tissue penetration of about 1 cm, allowing one to treat only superficial cancers [4]. Due to unlimited penetration through soft tissues, X-rays are considered the most preferable type of exciting irradiation for PSs in the treatment of deeply located tumors, and the corresponding method is referred to as X-PDT (X-ray-induced PDT) [5]. As the majority of sensitizers are not able to be excited directly by X-rays [6,7,8], agents for X-PDT usually consist of two individual components—scintillators (metal-based nano-particles) and PSs (organic molecules) [9]. Combining the properties of a scintillator and photosensitizer within one compound, the family of molybdenum and tungsten octahedral iodide clusters with the general formula [{M_6_I_8_}L_6_]^2−^ (M = Mo, W; L is an apical ligand) represent a unique “two in one” system [10,11,12]. Such clusters have a stable cluster core {M_6_I_8_}^4+^ that provides high photostability [13] and radiopacity due to the presence of heavy atoms [13]. Their generation of singlet oxygen under UV [14,15,16,17], white-light [18,19,20], and, most importantly, under X-rays [11,12] has also been proven, making them promising type II PSs for both conventional PDT and X-PDT. Despite the great prospects, to date there are only two studies demonstrating the X-ray-induced photodynamic effect of the clusters on living cells [11,12]. According to the literature, the main factors affecting radioluminescence efficiency are X-ray absorption and the quantum yield of photoluminescence [13]. Thus, among all studied compounds, {W_6_I_8_}^4+^ clusters, and in particular [{W_6_I_8_}I_6_]^2−^, were considered as the most efficient radioluminophores [13].

Unfortunately, the low hydrolytic stability of the clusters [21,22] greatly hinders their application in PDT and X-PDT. Incorporation into an organic matrix was proven to be an effective way of preserving cluster complexes in aqueous media [19,23,24,25,26]. In the context of PDT, a key requirement for the matrix is oxygen permeability. For example, luminescence of polystyrene cluster-containing particles was not sensitive to oxygen [23], while oxygen-permeable fluoropolymer films doped with clusters showed efficient photodynamic inactivation of bacteria and fungi [18,20]. A similar trend was observed for cluster-containing PLGA nanoparticles—the materials showed noticeable quenching in an oxygen-saturated environment, a high rate of ^1^O_2_ production, and pronounced phototoxicity against ovarian cancer cells under UV irradiation [27,28]. Another promising matrix is polyurethane (pU), whose oxygen permeability, as well as other physical properties, could be tailored by changing monomer type and ratio [29,30]. This polymer is biocompatible [31] and can be further modified [32,33], for example, with targeting molecules. Despite the methods of pU nanoparticle preparation being well-known, their application in biomedicine is still developing, as demonstrated by the few articles that have been published recently [34,35]. Concerning cluster-containing polyurethane, to date there have been two studies that have focused on bulk materials [36,37]. The authors demonstrated high response in terms of cluster luminescence in reaction to oxygen concentration in the atmosphere, confirming the prospect of using pU as a matrix for cluster complexes.

Here, using a simple nanoprecipitation technique, we have synthesized nanosized polyurethane particles doped with the most efficient radioluminescent cluster—[{W_6_I_8_}I_6_]^2−^—as a promising type II PS induced by both white light and X-rays. To incorporate clusters into pU chains, we synthesized and characterized the (choline)_2_[{W_6_I_8_}I_6_] complex, which contains an –OH group in its cationic part, before copolymerizing it with polyurethane monomers (polyethylene glycol, hexamethylene diisocyanate, and glycerin). For all materials obtained, stability, photo- and X-ray-induced luminescence, and oxygen sensing were studied, including determination of the Stern–Volmer constant, reversibility, and singlet oxygen generation efficiency under white light.

## 2. Materials and Methods

(K/Li)_2_[W_6_I_8_}I_6_] was synthesized according to procedure reported earlier [38]. All other reagents were purchased from Sigma Aldrich (St. Louis, MO, USA) and used as received.

The CHNS elemental analyses were performed using a Euro-Vector EA3000 elemental analyzer. Energy-dispersive X-ray spectroscopy (EDS) was performed on a Hitachi TM3000 TableTop SEM with Bruker QUANTAX 70 EDS equipment; results are reported as the ratio of the heavy elements: W and I. Tungsten content in samples was determined on a high-resolution spectrometer iCAP-6500 (Thermo Scientific, Waltham, MA, USA) with a cyclone-type spray chamber and “SeaSpray” nebulizer. Fourier-transform infrared spectroscopy (FTIR) spectra were recorded on a Bruker Vertex 80 as KBr disks. The absorption spectra were recorded on an Agilent Cary 60 UV/Vis spectrophotometer. Diffuse reflectance spectra of samples were recorded at room temperature on a Shimadzu UV-vis-near-IR 3101 PC spectrophotometer equipped with an integrating sphere. The spectra were converted to the Kubelka–Munk scale. The shape and morphology of particles were studied by transmission electron microscopy (TEM) on a Libra 120 microscope (Zeiss, Germany) at an acceleration voltage of 60 kV and by Dynamic light scattering (DLS) with Photocor Compact-Z equipment (Saint-Petersburg, Russia). Excitation and emission spectra of the samples in solid state and in solution were recorded using a Carry Eclipse (Agilent, Santa Clara, CL, USA).

### 2.1. Synthesis of (Choline)_2_[{W_6_I_8_}I_6_]

(K/Li)_2_[{W_6_I_8_}I_6_] (500 mg, 0.17 mmol, calculated for KLi[{W_6_I_8_}I_6_]) and choline iodide (choline is (CH_3_)_3_N(C_2_H_4_OH)^+^) (82.5 mg, 0.36 mmol) were dissolved in ethanol (100 mL). The solution was stirred for a day. Precipitated product was separated, washed with hot water twice, and dried at 50 °C in an oven. The powder obtained was dissolved in 5 mL of acetone, centrifuged to remove insoluble byproducts, and precipitated with a large excess of Et_2_O. (choline)_2_[{W_6_I_8_}I_6_] (1) was obtained with 76% yield (400 mg). For C_10_H_28_N_2_O_2_W_6_I_14_, the following composition was found: C, 4.0%; H, 1%; N, 0.9%. calcd: C, 3.9%; H, 0.9%; N, 0.9%. EDS: W/I = 6:14. IR (KBr), ν, cm^−1^: 3421 sh, 1627 m, 1466 s,1411 w, 1337 w, 1266 w, 1130 w, 1077 m, 1000 w, 952 s, 863 w.

### 2.2. Synthesis of Polyurethane Nanosized Particles

A solution of polyethylene glycol (PEG 600, 200 mg, 0.333 mmol) and hexamethylene diisocyanate (HDI, 74 mg, 0.44 mmol) in 4 mL of acetone was saturated with Ar. The dibutyltin dilaurate (DBTDL, 8 mg, 12.7 μmol) was then added, and the vial was tightly closed and heated at 70 °C for 20 min. After that, 1 mL of a solution of glycerin (4 mg, 43 μmol) and cluster complex 1 (0, 0.3 mg (0.01 μmol), 1.5 mg (0.05 μmol), 3 mg (0.1 μmol), or 15 (0.5 μmol)) in acetone was added to the reaction mixture, which was then heated at 70 °C for 100 min. The resulting mixture was added drop-wise to distilled water (V(H_2_O):V(Me_2_CO) = 10:1) under intensive stirring. The obtained dispersion was stirred for 10 min, filtered through a nylon filter (220 µm), and washed with water three times. The particles of 1^n^@pU (*n* = 0, 0.1, 0.5, 1, or 5, which corresponds to a mass percentage (%*w*/*w*) of 1 with respect to the mass of pU obtained at complete polymerization) were collected from the filter and dried at 40 °C. For the further investigations, dried particles were redispersed in distilled water at the required concentration.

### 2.3. Singlet Oxygen Generation

1,5-dihydroxynaphtalene (DHN), a well-known singlet oxygen trap, was used to determine singlet oxygen (^1^O_2_) generation efficiency. 1^0.1^@pU or 1^0.5^@pU particles (0.05 mg mL^−1^) were added to 20 mL of DHN solution in water (0.1 mM). The resulting colloid was sonicated and stirred for 5 min. The mixtures were irradiated with a spot light source L8253 (Hamamatsu) (400–800 nm, ~40 mW cm^−2^) with a UV 390 nm cut-off filter. At regular intervals of irradiation time (0, 1, 1.5, 2, 2.5, 3 min), 1 mL aliquots were collected. All aliquots were centrifuged, and UV-vis spectra of the supernatants were recorded. To determine ^1^O_2_ generation efficiency, absorption at 330 nm versus irradiation time was plotted and analyzed as a first-order reaction.

### 2.4. Stern–Volmer Dependence

A dispersion of 1^0^^.5^@pU particles (1 mg mL^−1^) in water was saturated with different ratios of O_2_/Ar gases (0, 21 (air), 33, 50, 67, and 100% oxygen). Before each experiment, the dispersion was bubbled with the gas mixtures for 3 min. This time was determined experimentally using pure O_2_ or Ar, and further saturation did not influence the luminescence intensity (λ_ex_ = 365 nm). K_SV_ was calculated according to S_(1/2)0_/S_(1/2)_ = 1 + K_SV_[O_2_], where S_(1/2)0_ and S_(1/2)_ signify integrated intensity of the left half of the spectra (500–645 nm) in the absence and presence of quencher, respectively, [O_2_] signifies the concentration of O_2_ in %, and K_SV_ represents the Stern–Volmer quenching constant.

### 2.5. XEOL

XEOL spectra of powdered samples (normalized on mole (in the case of pure clusters) or mass (in the case of polymers)) were recorded in ambient conditions on a home-built spectrometer [39] according to the procedure described in [13]. A CW X-ray tube BSV-27-Mo (Svetlana, St Petersburg, Russia) (40 kV × 20 mA) was used as the excitation irradiation source. The emission registration channel comprised a quartz optical imaging system, a grating monochromator (MDR-206, LOMO Photonics, St Petersburg, Russia, objective focus length 180 mm, grating 1200 lines per mm, inverse linear dispersion 4.3 nm mm^−1^) with input/output slits set to 2.2 mm/2.2 mm (spectral resolution about 10 nm), and a Hamamatsu H10493–012 photosensor module. Spectrum acquisition time was 18 min per single scan, and the spectra were averaged over 3 scans. XEOL spectra of 1^0.5^@pU, in the form of dispersion in D_2_O, were recorded as described in [40]. Subsequently, ~0.3 mL of material dispersion in D_2_O (100 mg mL^−1^) was placed in a cylindrical molybdenum glass ampoule, ultrasonicated, degassed by gentle bubbling with argon for 20 min, and sealed. The ampoule was placed in a cylindrical lead jacket possessing two 2-mm-wide vertical collimating slits at 90 degrees [41]. The estimated dose rate in air at the sample location was 85 krad h^−1^ [42].

## 3. Results and Discussion

### 3.1. Synthesis and Characterization

Generally, polyurethane (pU) polymers are synthesized by the reaction of OH– and NCO-containing monomers with the formation of a key polyurethane group, –N(H)–C(O)O– (Appendix A). The polymerization reaction occurs slowly at room temperature and could be catalyzed by strong bases (e.g., dibutyltin dilaurate (DBTDL) [43,44] or lithium tert-butoxide [45,46]), usually under moderate heating. For the active component of the material, a [{W_6_I_8_}I_6_]^2−^ tungsten cluster was chosen due to better X-ray absorption leading to brighter XEOL in comparison to molybdenum analogues [11,13,21]. To impregnate the polyurethane matrix with a cluster complex, the new compound, (choline)_2_[{W_6_I_8_}I_6_] (choline—(2-hydroxyethyl)trimethylammonium (Appendix A)), with two OH groups was synthesized. (choline)_2_[{W_6_I_8_}I_6_] (1) was obtained with high yield by the reaction of cation metathesis from (K/Li)_2_[{W_6_I_8_}I_6_] and choline iodide. Structure and composition were proven using CHN, EDS, FTIR (Appendix A), and ^1^H-NMR analysis (Appendix A). The complex obtained can take part in the polymerization process as a functional monomer. Thus, a schematic representation of the synthetic techniques, i.e., illustration of cluster impregnation into the polyurethane matrix and nanoparticles formation, is presented in Figure 1.

The nanoprecipitation technique, i.e., drop-by-drop addition of the reaction mixture into a great excess of vigorously stirred distilled water, was chosen to obtain polymeric cluster-containing nanoparticles. Therefore, it was essential to obtain a solution of the cluster-containing polymer as a result of the reaction. For example, addition of more than 5% of DBTDL (of monomer) or reducing solvent volume to 2 mL resulted in a more complete reaction and, therefore, in the precipitation of an insoluble in acetone long-chain polymer. Thus, in order to use the nanoprecipitation technique we were forced to diminish polymerization completeness so as to obtain a solution of polymer, which resulted decreased reaction yield. The amount of DBTDL and the acetone volume were set at 3% and 4 mL, respectively. To study the effect of cluster concentration on polymerization, the reaction was conducted in the presence of different amounts of (choline)_2_[{W_6_I_8_}I_6_] (0, 0.1, 0.5, 1, and 5%w/w of cluster, which corresponds to 0, 1, 5, 10, and 50 mg of cluster per 1 g of final polymer, provided 100% efficacy of polymerization). The samples obtained were labelled as pU and 1^n^@pU (*n* = 0.1, 0.5, 1, 5). The 5% concentration led to precipitation of the cluster from the reaction mixture, and thus 1^5^@pU was not obtained. In the case of other concentrations, slightly cloudy viscous solutions were obtained and used for nanoprecipitation.

According to TEM (Figure 2a–c), all of the samples consisted of spherical nanoparticles with d = 42 ± 12 nm (pU), d = 36 ± 13 nm (1^0.1^@pU), and d = 28 ± 7 nm (1^0.5^@pU) (Figure 2d–f), indicating the cluster had a slight effect on the polymerization process and particle morphology. However, in 1^1^@pU, along with polymeric particles, contrasting agglomerates of clustered nanocrystals were observed (Appendix A), due to which this sample was excluded from further studies. DLS analysis showed hydrodynamic diameters of ~160 nm with close to zero ζ-potentials for all samples, evidencing the formation of small aggregates in the solution. Successful polymerization was proven using FTIR spectroscopy (Figure 2g). Samples show the absence of a characteristic peak in the –NCO group from the initial HDI at approximately 2270 cm^–1^ [47]. At the same time, there is a peak in N–H bonds at 3335 cm^–1^, which is intrinsic for polyurethane [47]. Additionally, there are characteristic peaks for such bonds as C–H (~2930, 2856, ~1580, and ~1470 cm^−1^), C=O (1621 cm^−1^), C–N (~1258 cm^−1^), and C–O–C (1180 cm^−1^) [48]. The precise content of tungsten in 1^0.5^@pU was determined using ICP-AES and recalculated as mass of 1 per 1 g of pU. Thus, during the reaction, ~6.7 mg of the cluster is incorporated into 1 g of polymer, which is ~30% of the overall cluster in the reaction mixture.

Absorbance of the samples was studied via diffuse reflectance spectroscopy (DRS) using the Kubelka–Munk function (Figure 2h). Pure pU is mostly absorbed in the UV region, with a shoulder that prolongs up to ~450 nm. In turn, 1^n^@pU (*n* = 0.1, 0.5) demonstrates cluster-related absorption up to ~550 nm, with a pronounced dependence of the absorption intensity on n, i.e., the higher the cluster content, the higher the absorption intensity (Figure 2h). In order to study the stability of cluster-containing materials, 1^0.5^@pU was soaked in water for 7 days and studied using DRS (Figure 2h). According to the data obtained, soaked samples demonstrated an identical profile of absorption spectra as fresh samples, indicating superb stability in aqueous media.

### 3.2. Photoluminescence

The emission spectrum profile of 1 in acetonitrile is almost similar to the profile of the well-known (Bu_4_N)_2_[{W_6_I_8_}I_6_], with emission maxima (λ_max_) at ~650 nm (Figure 3a) indicating preservation of cluster core composition and ligand environment during the metathesis reaction. In a solid state, structural characteristics and, in particular, molecular packing have a strong effect on luminescence parameters, such as the shape of the spectra, λ_max_, and emission intensity. Diminishing distances between cluster units usually results in luminescence quenching and the bathochromic shift of λ_max_, which was observed in the case of clusters with small counterions [11,49,50]. Indeed, the emission maximum for cluster 1 is redshifted by 15 nm (650 nm vs. 635 nm), and FWHM (full width at half maximum) is 30 nm larger (147 nm vs. 117 nm) than that of (Bu_4_N)_2_[{W_6_I_8_}I_6_]. Much weaker emission of 1 in comparison to (Bu_4_N)_2_[{W_6_I_8_}I_6_] (Appendix A) can be seen, even with the naked eye. Interestingly, incorporation of 1 in the polyurethane matrix dramatically increases luminescent intensity while preserving the emission maximum at 645 nm (Figure 3b), indicating a decrease in the influence of the cation on cluster luminescence [38]. Additionally, the pronounced dependence of emission intensity on cluster content was observed (Figure 3b). Moreover, cluster emission can be additionally increased due to reabsorption of light emitted from excited polyurethane as a result of overlap with the excitation spectra of 1^n^@pU (Appendix A). This suggestion is confirmed by the gradual decrease in polymer emission at ~410 nm coinciding with the increase in n.

### 3.3. O_2_ Generation

Molecular O_2_ is known to be one of the main quenchers of cluster emission, and interaction of triplet oxygen with excited cluster units results in the formation of ^1^O_2_ [14,15,19,51]. Here, to determine ^1^O_2_ generation efficiency, we used a selective trap for singlet oxygen—1,5-dihydroxynaphthalene (DHN). The oxidation of DHN by ^1^O_2_, with the formation of juglone, could be detected by UV-vis spectroscopy. The aqueous dispersions of 1^0.1^@pU and 1^0.5^@pU (0.05 mg mL^−1^) in the presence of DHN were irradiated with white light for several minutes. Characteristic spectra are given in ESI (Appendix A). Linear approximation of ln(A/A_0_) at 330 nm vs. time provides effective rate constant values (k_eff_) (Figure 4). One can see that, similar to emission intensity, the efficiency of ^1^O_2_ production strongly depends on the cluster content in the matrix—k_eff_ is equal to 0.01 min^−1^ and 0.03 min^−1^ for 1^0.1^@pU and 1^0.5^@pU, respectively. Since there is no information on rate constants of ^1^O_2_ production by tungsten clusters or cluster-containing materials in water or, importantly, under white light, only indirect comparisons of effectiveness can be made. Nevertheless, k_eff_ of 1^0.5^@pU (C_1_ = 1.1 × 10^–5^ M, concentration of 1 was calculated from ICP-AES data) is comparable to k_eff_ of true solutions of pure tungsten clusters in CH_3_CN, such as (Bu_4_N)_2_[{W_6_I_8_}I_6_] (C = 2 × 10^−5^ M, λ_ex_ = 365 nm, k_eff_ = 0.049 min^−1^) or (Bu_4_N)_2_[{W_6_I_8_}(SCN)_6_] (C = 2 × 10^–5^ M, λ_ex_ = 365 nm, 0.059 min^–1^) [17]. Note that, due to the low lifetime of singlet oxygen in H_2_O (about ~3.5 μs) [52] and less effective excitation of clusters with white light, we can conclude that 1^0.5^@pU is one of the most efficient cluster-based photosensitizers.

### 3.4. Oxygen Sensing and Stability in Water

Cluster emission intensity is highly sensitive to molecular oxygen, and quenching occurs due to energy transfer to ^3^O_2_ with the formation of ^1^O_2_ [53,54]. When quenchers play a key role, processes could be characterized by the Stern–Volmer constant (K_SV_). Here, we studied the luminescence response of an aqueous dispersion of 1^0.5^@pU particles (0.1 mg mL^−1^) to different ratios of O_2_/Ar gases (0, 21 (air), 33, 50, 67, and 100% oxygen) (Figure 5a). Due to the low sensitivity of some photodetectors in far red and NIR regions, integrated intensity of the left part of the spectra (up to 645 nm) was used for estimation of the Stern–Volmer constant. Thus, the linear fit of a plot of S_(1/2)0_/S_(1/2)_ (S_(1/2)0_ is integrated intensity of the left part of spectra up to 645 nm in the absence of oxygen) vs. oxygen concentration yields a K_SV_ value of 0.042%^−1^. The obtained value is comparable to those of other cluster-containing materials, e.g., antibacterial films based on fluoropolymer demonstrated K_SV_ values of 0.075%^–1^ in the case of (Bu_4_N)_2_[{Mo_6_I_8_}(CF_3_(CF_2_)_5_COO)_6_] [18] and of 0.013%^−1^ in the case of (Bu_4_N)_2_[{Mo_6_I_8_}(OTs)_6_] (OTs^−^ is *p*-toluenesulfonate) [20]. The linear shape of the Stern–Volmer plot confirms that the quenching by oxygen was not limited by diffusion [55].

Stability of the material was also studied in cycling experiments (Figure 5b). Luminescence spectra of fresh 1^0.5^@pU and 1^0.5^@pU soaked for 7 days were recorded in air (first point) and under pure Ar or O_2_ conditions (five cycles each). Normalized integrated intensities of the left part of the spectrum (up to 645 nm) under different atmospheres show that the quenching is reversible. In Ar atmosphere, the mean value is 97 ± 2%, and in oxygen it is 16.4 ± 0.4%, indicating the quenching of ~80% of excited cluster units. Particles soaked in water for 7 days showed a similar tendency, despite maximal emission in Ar atmosphere slightly diminishing to 86 ± 3%. We associate this with some form of aggregation process, since intensities with air and O_2_ were equal to those of the fresh sample. The aggregation probably could be overcome by using more intense ultrasonication. DLS data for diluted soaked samples show results similar to those of the fresh sample, with a d value of ~160 nm and close to zero ζ-potential also confirming preservation of the particles in solution. Thus, the data obtained confirm that the polyurethane matrix effectively prevents hydrolysis of the cluster and, due to high oxygen permeability, preserves oxygen sensitivity.

### 3.5. X-ray-Excited Optical Luminescence (XEOL)

Along with photoinduced luminescence, cluster complexes are known to possess X-ray-induced optical luminescence in both solids and solutions [10,13,40,56]. Here, we studied XEOL for powdered 1 and 1^n^@pU (*n* = 0, 0.1, 0.5) and dispersions of 1^0.5^@pU in D_2_O, since radioluminescence is strongly quenched in H_2_O [57]. (Bu_4_N)_2_[{W_6_I_8_}I_6_] [13] was used as an internal control and, in the case of pure clusters, the spectra obtained were normalized to the number of moles. Similar to photoluminescence, the cation had a noticeable effect on emission intensity and the position of the emission maximum for powdered clusters (Figure 6a); substituting Bu_4_N^+^ with choline^+^ leads to around a 65-fold decrease in luminescence intensity and the bathochromic shift of λ_max_ by 40 nm (675 vs. 715 nm for (Bu_4_N)_2_[{W_6_I_8_}I_6_] and 1, respectively). This indicates similar mechanisms for both photo- and X-ray-induced luminescence [10,13]. Pure pU and 1^0.1^@pU (Figure 6b), having low cluster content, do not show any detectable luminescence under X-rays. In turn, 1^0.5^@pU, containing more clusters, demonstrates noticeable emissions at an λ_max_ of 690 nm. So far, as emission intensity is related to the mole of the emitter, the emission intensity of 1^0.5^@pU is ~15 times higher than if it were calculated using the ratio of real cluster content (6.7 mg in 1 g of sample) to the mass of 1. Overall, this again confirms a decrease in the influence of the cation on cluster luminescence after its inclusion in polymer.

Since a gradual decrease in emission intensity was observed during the experiment, XEOL spectra under prolonged X-ray irradiation were recorded for (Bu_4_N)_2_[{W_6_I_8_}I_6_], 1, and 1^0.5^@pU in a solid state. Three averaged raw spectra of three consecutive scans (18 min each) are presented in Figure 7a–c. The spectra were smoothed and the rate constants and degree of decomposition were calculated from the I_0_/I vs. irradiation time plot (Table 1, Appendix A). It can be seen that the emission intensity of (Bu_4_N)_2_[{W_6_I_8_}I_6_] under prolonged X-ray irradiation decreases by 6% within ~1 h (second set of three scans) and by 11% after ~2 h (third set of three scans) (Figure 7a, Table 1). On the other hand, in similar conditions, the intensity of complex 1 decreases by 32 and 53% (Figure 7b, Table 1). Cluster 1 being incorporated into the polyurethane matrix, i.e., 1^0.5^@pU, demonstrates higher stability, with intensity loss of 8 and 16% for the second and third set of scans, respectively (Figure 7c, Table 1).

According to the results of our previous work [40], a high concentration of the luminescent component is required to study X-ray luminescence in water or D_2_O. It is also necessary to saturate the solution/dispersion with an inert gas to remove oxygen from the system and, as a result, reduce luminescence quenching. Here, dispersions of neat pU and 1^0.5^@pU in D_2_O at a concentration of 100 mg mL^−1^ were prepared in a molybdenum glass tube and bubbled with Ar gas. Molybdenum glass was used due to low parasitic emission [40]. Emission spectrum of a tube with pure D_2_O was used as a baseline (Appendix A). It was shown that neat polyurethane particles do not exhibit any radioluminescence (Appendix A). In turn, despite low cluster concentration, 1^0.5^@pU demonstrates noticeable emission under X-rays in D_2_O (Figure 7d) without any loss in intensity during ~3 h of irradiation, indicating preservation of the cluster in the aqueous dispersion. To the best of our knowledge, this is the first evidence of XEOL with polyurethane-based nanoparticles in water.

## 4. Conclusions

Herein, cluster complex (choline)_2_[{W_6_I_8_}I_6_] (1), containing two –OH groups, was synthesized, characterized by CHN, EDS, and ^1^H-NMR, and successfully copolymerized with monomers to obtain polyurethane polymer. To obtain nanoparticles, we used a simple nanoprecipitation technique, i.e., the addition of the reaction mixture to water under intensive stirring. As a result, highly luminescent pU particles with diameter of 30–40 nm doped with [{W_6_I_8_}I_6_]^2−^ were obtained. The effect of cluster concentration on morphology and photophysical properties was studied. It was shown that cluster-containing particles possess bright photoluminescence and superb efficiency in singlet oxygen generation in aqueous dispersions under white light. Furthermore, it was shown that luminescence intensity changed quickly in relation to oxygen concentration in water. More importantly, X-ray-excited optical luminescence was detected both for powdered samples and for dispersions of materials in D_2_O. The emission intensity was preserved even after ~3 h of irradiation in D_2_O, indicating the high stabilizing efficiency of the pU matrix. To the best of our knowledge, we have demonstrated the XEOL of polyurethane-based nanoparticles in aqueous media for the first time. Thus, the set of properties that the obtained materials possess makes them promising for biomedical applications, especially as photo and X-ray sensitizers for deep photodynamic therapy.

## Figures and Tables

**Figure 1 nanomaterials-12-03580-f001:**
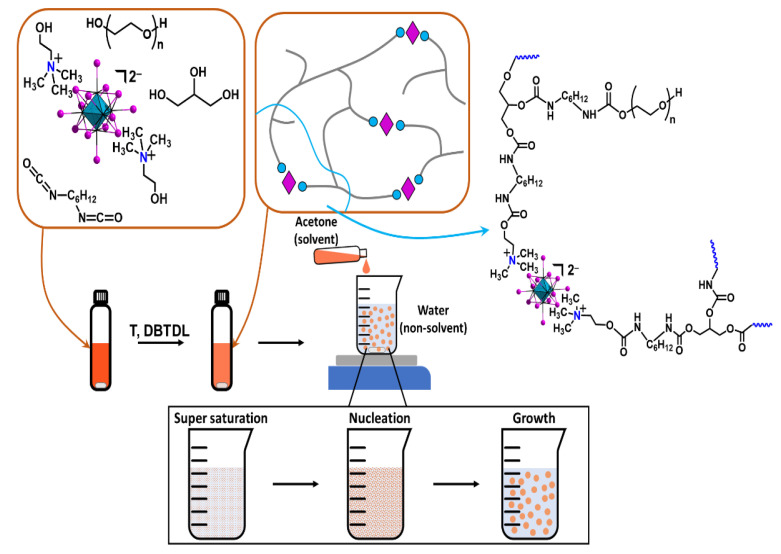
Schematic representation of polyurethane polymerization with a cluster and the formation of nanoparticles.

**Figure 2 nanomaterials-12-03580-f002:**
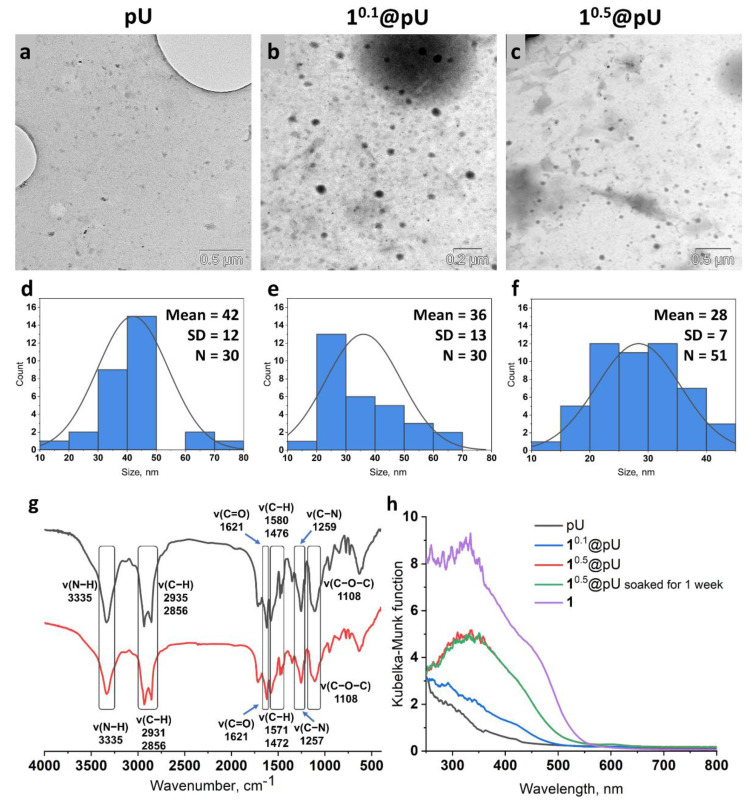
(**a**–**c**) TEM pictures of pU (scale 0.5 µm), 1^0.1^@pU (scale 0.2 µm), and 1^0.5^@pU (scale 0.5 µm); (**d–f**) statistics for size distribution from the TEM pictures; (**g**) FTIR spectra for neat pU particles and for the sample of 1^0.5^@pU; (**h**) diffuse reflectance spectra of pure pU, 1^0.1^@pU, 1^0.5^@pU, 1^0.5^@pU soaked in water for one week, and cluster 1 converted to absorption spectra using the Kubelka–Munk function.

**Figure 3 nanomaterials-12-03580-f003:**
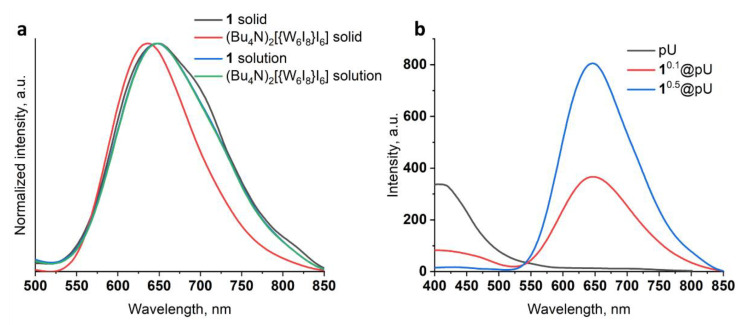
(**a**) Normalized luminescence spectra of (Bu_4_N)_2_[{W_6_I_8_}I_6_] and 1 in acetonitrile solution and in a solid state; (**b**) luminescence spectra of 1, 1^0.1^@pU, and 1^0.5^@pU in a solid state.

**Figure 4 nanomaterials-12-03580-f004:**
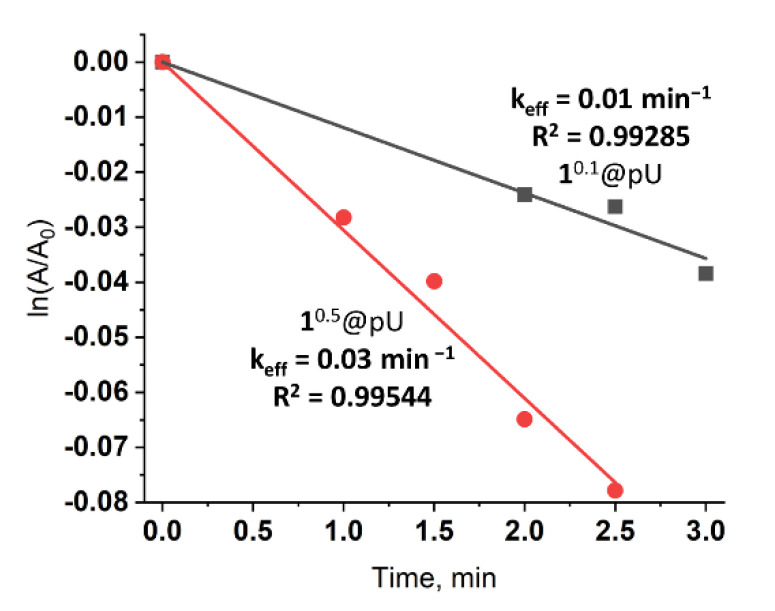
ln(I/I_0_) vs. time plots for the reaction of DHN (0.1 mM) transformation in the presence (0.05 mg mL^−1^) of 1^0.1^@pU (grey squares) and 1^0.5^@pU (red circles) in water (20 mL). The line of the corresponding color represents linear approximation.

**Figure 5 nanomaterials-12-03580-f005:**
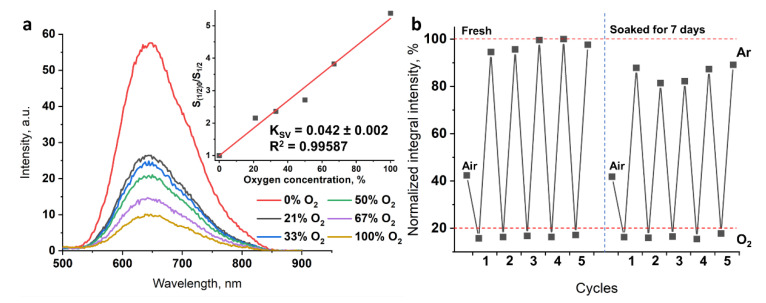
(**a**) Luminescence spectra of aqueous solutions of 1^0.5^@pU in different mixtures of O_2_/Ar atmosphere and under air conditions (21% of oxygen). The insert is a plot of S_(1/2)0_/S_(1/2)_ vs. oxygen concentration and its linear approximation. (**b**) The value of integrated luminescence intensity for the left part of the spectra up to 645 nm under air, 100% O_2_, and 100% Ar for fresh and soaked samples. At each point the solution was saturated with gases for 3 min.

**Figure 6 nanomaterials-12-03580-f006:**
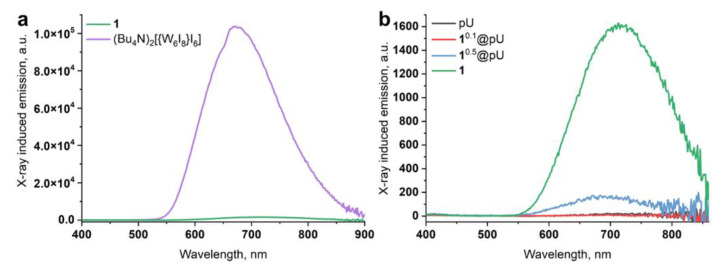
XEOL spectra for: (**a**) cluster 1 and (Bu_4_N)_2_[{W_6_I_8_}I_6_]; and (**b**) cluster 1, pure pU nanoparticles, and 1^0.1^@pU and 1^0.5^@pU samples. Each line is an average from three scans lasting 18 min each. Cluster 1 is added to both pictures to facilitate the comparison.

**Figure 7 nanomaterials-12-03580-f007:**
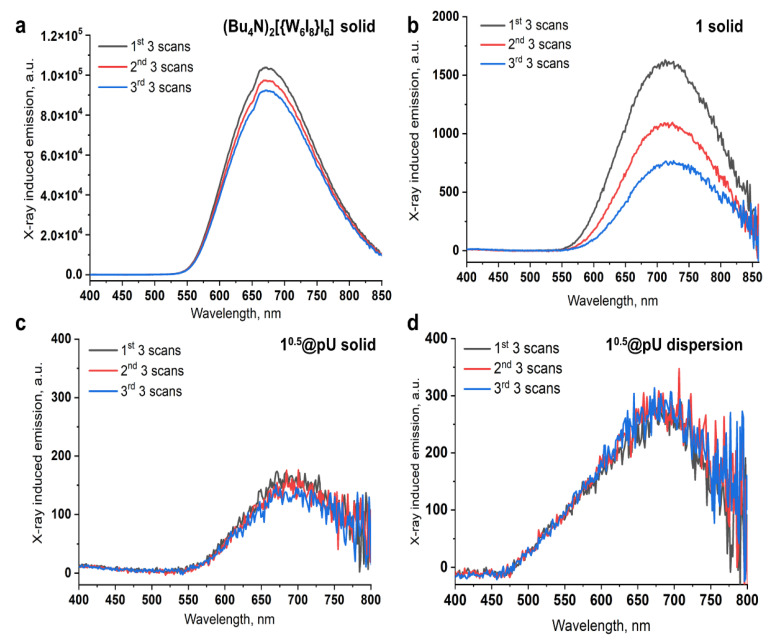
XEOL spectra during 3 h of irradiation for (**a**) (Bu_4_N)_2_[{W_6_I_8_}I_6_], (**b**) 1, and (**c**) 1^0.5^@pU in solids and for (**d**) the D_2_O dispersion of 1^0.5^@pU. Each line is an average of three scans lasting 18 min each. Total time of irradiation was ~3 h.

**Table 1 nanomaterials-12-03580-t001:** Degradation of XEOL intensity under prolonged X-ray irradiation.

Sample	Intensity Loss, %		Rate Constant, min^−1^
	2nd 3 Scans	3rd 3 Scans	
(Bu_4_N)_2_[{W_6_I_8_}I_6_]	6%	11%	0.0011
1	32%	53%	0.01
1^0.5^@pU solid	8%	16%	0.0019
1^0.5^@pU in D_2_O	0%	0%	-

## Data Availability

The data presented in this study are available on request from the corresponding author.

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
