# Peer review of "Oxygen-Sensitive Photo- and Radioluminescent Polyurethane Nanoparticles Modified with Octahedral Iodide Tungsten Clusters"

_nanomaterials, 2022, doi:10.3390/nano12203580_

Round 1
Reviewer 1 Report
Vyacheslav A. Bardin and coworkers report the nanosized polyurethane (pU) particles containing octahedral iodide tungsten clusters-(choline)2[{W6I8}I6] (1). The 1n@pU nanoparticles obtained here achieves singlet oxygen generation induced by both white light and X-rays. Also, as promising candidate for X-ray-induced photosensitizers, this cluster complexes address the key problems: 1) improving the hydrolytic stability of molybdenum clusters; 2) advancing the oxygen permeability of matrix (pU); 3) achieving better X-rays absorption leading to brighter X-ray excited optical luminescence (XEOL) in aqueous dispersion. Most of the conclusions are supported by scientific evidence. Nonetheless, I have some remarks that I am listing here.
1. The introduction is a little bit tedious. In the first paragraph, the author addressed the concept of X-ray-induced photodynamic therapy (X-PDT), yet this work involved no biological or therapeutic experiment. Therefore, shorten the background of cancer and brief explain of concepts (PDT, photosensitizers and X-PDT) would be appropriate. Also, in the following paragraph, further explanation on the material design (e.g., modulation of halogen atoms-iodine in W6 clusters which may improve the X-ray absorption ability) is welcomed.
2. The discussion about the absorption in line 234, 235 (“a pronounced dependence of the absorption intensity on n” where n = 0.1 and 0.5) and the excitation in line 265 (“the pronounced dependence of the emission intensity on cluster content was observed”) were given with uncertainty. Considering the sample 1n@pU (n =1, 5) were excluded from this work, the speculative conclusion from these spectrums (“the higher the cluster content the higher the absorption intensity” and “the higher the content of 1 the lower the emission of the polymer”) should be consolidated with more data.
3. In the characterization of photoluminescence, the samples are dissolved in acetonitrile solution, whereas in other characterization (1O2 generation, oxygen stability and et al.), expect the solid samples, the cluster complexes are mostly dispersed in water or D2O. Based on the consideration that “extremely low concentration of the material in solution, low content of the complex in the matrix” and the inherent low hydrolytic stability of molybdenum clusters, further discussions and additional references should be described to clarify this point.
4. The discussions about XEOL need recheck. The explanation of Figure 7b (“intensity decreases by ~30%”) is not stringent enough. Author should better calculate the decrease rate and consolidate this interpretation after fitting the data. Also, the description of Figure 7c is inaccurate (“no noticeable changes were observed” of 10.5@pU). In Figure 7d, the explanation (“without any loss in the intensity”) is ambiguous. The sharp peaks interfere the judgment of results.
5. To improve this work, several details are required to be supplemented: The legend of Figure 6 was omitted; In the supporting information and the corresponding part in main text, all legends of the Figures were written as 1S, 2S, …; Meanwhile, some expressions are ambiguous and hard to read. Preferably, the phrasing requires revise: for example, in line 350, “confirms a decrease in the influence of the cation on cluster luminescence after its inclusion in polymer”;
Author Response
Vyacheslav A. Bardin and coworkers report the nanosized polyurethane (pU) particles containing octahedral iodide tungsten clusters-(choline)2[{W6I8}I6] (1). The 1n@pU nanoparticles obtained here achieves singlet oxygen generation induced by both white light and X-rays. Also, as promising candidate for X-ray-induced photosensitizers, this cluster complexes address the key problems: 1) improving the hydrolytic stability of molybdenum clusters; 2) advancing the oxygen permeability of matrix (pU); 3) achieving better X-rays absorption leading to brighter X-ray excited optical luminescence (XEOL) in aqueous dispersion. Most of the conclusions are supported by scientific evidence. Nonetheless, I have some remarks that I am listing here.
Answer: Thank you for your positive feedback and relevant suggestions, which we address below.
- The introduction is a little bit tedious. In the first paragraph, the author addressed the concept of X-ray-induced photodynamic therapy (X-PDT), yet this work involved no biological or therapeutic experiment. Therefore, shorten the background of cancer and brief explain of concepts (PDT, photosensitizers and X-PDT) would be appropriate. Also, in the following paragraph, further explanation on the material design (e.g., modulation of halogen atoms-iodine in W6clusters which may improve the X-ray absorption ability) is welcomed.
Answer. The introduction part was rewritten in accordance to the reviewer’s comment.
- The discussion about the absorption in line 234, 235 (“a pronounced dependence of the absorption intensity on n” where n = 0.1 and 0.5) and the excitation in line 265 (“the pronounced dependence of the emission intensity on cluster content was observed”) were given with uncertainty. Considering the sample 1n@pU (n =1, 5) were excluded from this work, the speculative conclusion from these spectrums (“the higher the cluster content the higher the absorption intensity” and “the higher the content of 1 the lower the emission of the polymer”) should be consolidated with more data.
Answer. Here, we do not discuss quantitative dependency of absorption and emission on the cluster concentration. Taking into account similar particle size of cluster-containing materials, measurements of absorption and emission in solid state should give reliable data at least for qualitative comparison. Thus, all of the data obtained, i.e., absorption, photo and radio-luminescence, are in agreement with the suggestion “the higher the cluster content the higher absorption/emission intensity”. In addition, no XEOL emission was observed for pure pU and 10.1@pU, which is also confirms strong dependency of the optical properties on cluster content in polyurethane matrix.
- In the characterization of photoluminescence, the samples are dissolved in acetonitrile solution, whereas in other characterization (1O2 generation, oxygen stability and et al.), expect the solid samples, the cluster complexes are mostly dispersed in water or D2O. Based on the consideration that “extremely low concentration of the material in solution, low content of the complex in the matrix” and the inherent low hydrolytic stability of molybdenum clusters, further discussions and additional references should be described to clarify this point.
Answer. Only pure clusters were dissolved in acetonitrile since they are insoluble in water. Polyurethane materials were studied in water/D2O dispersion (see 3.4 Oxygen sensing and stability in water) and in solid state (3.2 Photoluminescence). We tried to conduct the experiments on the determination of singlet oxygen production efficiency in acetonitrile, but unfortunately, 1n@pU nanoparticles do not form stable dispersion in this solvent. Since in our manuscript no comparisons between pure clusters and cluster-containing materials apart from emission maxima were made, we believe, that the correlations obtained are reliable. Corresponding section was rephrased for clarity.
- The discussions about XEOL need recheck. The explanation of Figure 7b (“intensity decreases by ~30%”) is not stringent enough. Author should better calculate the decrease rate and consolidate this interpretation after fitting the data. Also, the description of Figure 7c is inaccurate (“no noticeable changes were observed” of 10.5@pU). In Figure 7d, the explanation (“without any loss in the intensity”) is ambiguous. The sharp peaks interfere the judgment of results.
Answer. Thank you for your valuable comment. Indeed, due to the high noise level, it is impossible to correctly determine the rate and degree of decomposition of complexes and materials. The spectra were smoothed and from the dependency of I0/I on time rate constants and degree of decomposition were calculated (See attached file). No noticeable changes were observed for the dispersion of the material in D2O, but in the case of powdered sample degradation rate was found to be similar to (Bu4N)2[{W6I8}I6]. Corresponding discussion was added to the main text. Also, Figure 7b and 7c were mixed up.
- To improve this work, several details are required to be supplemented:
The legend of Figure 6 was omitted;
Answer. Figure 6 was modified.
In the supporting information and the corresponding part in main text, all legends of the Figures were written as 1S, 2S, …;
Answer. All captions of the Figures in main text and in ESI were renamed as S1, S2…
Meanwhile, some expressions are ambiguous and hard to read. Preferably, the phrasing requires revise: for example, in line 350, “confirms a decrease in the influence of the cation on cluster luminescence after its inclusion in polymer”
Answer. Text of the article was edited and ambiguous expressions were rephrased.

Reviewer 2 Report
By using simple nanoprecipitation technique, the authors have synthesized nanosized pol-82 yurethane particles, doped with octahedral halide tungsten cluster, as promising third 83 generation PSs of type II induced by both white light and X-rays. Following issues can be addressed to improve the manuscript.
1. How to form the nanoparticles? The authors should show the formation of nanoparticles in Figure 1.
2. In Figure 2d-f, please revise "length" to "size".
3. The authors can try to calculate the 1O2 generating efficacy.
4. How about the limit of detection for oxygen sensing.
Author Response
By using simple nanoprecipitation technique, the authors have synthesized nanosized polyurethane particles, doped with octahedral halide tungsten cluster, as promising third generation PSs of type II induced by both white light and X-rays. Following issues can be addressed to improve the manuscript.
Answer: Thank you for your positive feedback and relevant suggestions, which we address below.
- How to form the nanoparticles? The authors should show the formation of nanoparticles in Figure 1.
Answer. The Figure 1 was supplemented with a scheme for the growth of nanoparticles.
- In Figure 2d-f, please revise "length" to "size".
Answer. Corrected.
- The authors can try to calculate the 1O2 generating efficacy.
Answer. Thank you for your valuable comment. The best way to measure 1O2 generating efficacy is the direct measurement of singlet oxygen quantum yield (ΦΔ) by its emission. Unfortunately, we have no direct access to such equipment. Estimating of relative singlet oxygen quantum yield (ΦΔ) by comparison with other photosensitizers (for example, [Ru(bpy)3]2+) is also a good way to benchmark photosensitizing capacity of an agent. Nevertheless, the reference molecules, such as (Bu4N)2[{Mo6I8}I6] or [Ru(bpy)3]2+, would be used in the form of a true solution, and our materials are obtained in the form of dispersion of particles of fixed size and surface area. Thus, we believe that such comparison would be highly unreliable in this study.
- How about the limit of detection for oxygen sensing.
Answer. The measurements carried out in this work were made with hand-made instruments, i.e., by simple mixing fixed volumes of gases (1/1, 1/2, 2/1). Measurement of the limit of oxygen detection requires precise equipment that we do not have direct access to.

Reviewer 3 Report
In the present manuscript, the authors describe a polyurethane nanoparticle modified with octahedral iodide tungsten cluster to be applied in cancer therapy. However, authors should review the manuscript according to the following comments:
1 - The novelty of the article is not clearly presented. Further, the authors have to do more assays to prove the potential of the work.
2. The introduction section of the manuscript is incomplete. The authors must add more examples available in the literature and clarify the work and its importance.
3. The authors used tungsten as a compound to synthesize the clusters. This choice over other materials should be explained and some advantages of tungsten over the other materials used to perform PDT should also be described.
4. The text should be carefully checked, modified, and ensured uniform format throughout the article, such as, “set at 3 % and” (line 212) and “The concentration of 5% led” (line 217), and so on.
5. In section 3.1, the nanoparticles’ surface charge should be determined.
6. The author must evaluate the cell viability of normal and cancer cells after incubation with the system to demonstrate its biocompatibility.
7. Further, the analysis of the cytotoxicity effect of the system when incubated in cancer cells must be evaluated to demonstrate the cancer cell death mediated by irradiation.
8. In section 3.4, the authors evaluate the stability of the nanoparticles in water by cycling experiments. In reviwer's opinion authors must analyze the stability of the nanoparticles after dispersion in water (pH=7.4) for different periods by measuring the size, and check for any nanoparticles aggregation.
Author Response
In the present manuscript, the authors describe a polyurethane nanoparticle modified with octahedral iodide tungsten cluster to be applied in cancer therapy. However, authors should review the manuscript according to the following comments:
Answer: Thank you for your positive feedback and relevant suggestions, which we address below.
- The novelty of the article is not clearly presented. Further, the authors have to do more assays to prove the potential of the work.
- The introduction section of the manuscript is incomplete. The authors must add more examples available in the literature and clarify the work and its importance.
- The authors used tungsten as a compound to synthesize the clusters. This choice over other materials should be explained and some advantages of tungsten over the other materials used to perform PDT should also be described.
Answer (1-3). The introduction part was rewritten. Additional references were added. The choice of tungsten cluster was explained.
- The text should be carefully checked, modified, and ensured uniform format throughout the article, such as, “set at 3 % and” (line 212) and “The concentration of 5% led” (line 217), and so on.
Answer. Corrected.
- In section 3.1, the nanoparticles’ surface charge should be determined.
Answer. Close to zero ζ-potential was observed for fresh and soaked for 1 week samples. Corresponding discussion was added in the main text.
- The author must evaluate the cell viability of normal and cancer cells after incubation with the system to demonstrate its biocompatibility.
- Further, the analysis of the cytotoxicity effect of the system when incubated in cancer cells must be evaluated to demonstrate the cancer cell death mediated by irradiation.
Answer (6-7). Thank you for your valuable comment. Indeed, we are keen on further biological studies, including dark cytotoxicity and photoinduced cytotoxicity under white light and X-rays, cellular uptake and elimination, surface modification with antibodies, etc, and this work is currently under the way. In our manuscript we demonstrated stability of the material and physicochemical properties which could be useful for biological applications. Thus, we believe, that this data can be excessive here and would be more appropriate for further fully bio-related article.
- In section 3.4, the authors evaluate the stability of the nanoparticles in water by cycling experiments. In reviwer's opinion authors must analyze the stability of the nanoparticles after dispersion in water (pH=7.4) for different periods by measuring the size, and check for any nanoparticles aggregation.
Answer. We measured size and ζ-potential for fresh and soaked for 1 week samples (pU and 10.5@pU). According to the data obtained, both fresh and aged samples consist of small aggregates with diameter of ~160 nm having close to zero ζ-potential, which indicates high colloidal stability of the materials. Corresponding discussion was added in the main text. The dependency of stability on pH will be studied in further bio-related article.
